# Improvement in the Management of Oral Anticoagulation in Patients with Atrial Fibrillation in Primary Health Care

**DOI:** 10.3390/ijerph19116746

**Published:** 2022-05-31

**Authors:** África García-Roy, Ana Sarsa-Gómez, Fátima Méndez-López, Blanca Urdin-Muñoz, María Antonia Sánchez-Calavera

**Affiliations:** 1Las Fuentes North Health Center, Aragon Health Service, 50002 Zaragoza, Spain; africag.roy@gmail.com (Á.G.-R.); amsarsa@gmail.com (A.S.-G.); blurdin@gmail.com (B.U.-M.); mascalavera62@hotmail.com (M.A.S.-C.); 2Aragonese Primary Care Research Group (GAIAP), Aragon Health Research Institute (IISA), 50009 Saragossa, Spain; 3Department of Internal Medicine, Psychiatry and Dermatology, University of Zaragoza, 50009 Zaragoza, Spain

**Keywords:** atrial fibrillation, oral anticoagulation, anticoagulation management, drug monitoring

## Abstract

(1) Background: Evaluation and improvement of the management of patients with atrial fibrillation in treatment with oral anticoagulants from primary health care. (2) Methods: prospective quasi-experimental study, conducted on 385 patients assisted with Atrial Fibrillation (AF) at the Las Fuentes Norte Health Center, before and after the implementation of actions to improve oral anticoagulants management from October 2015 to July 2017. (3) Results: The ACO-ZAR I study revealed that the population with AF presents a global prevalence of 1.7%, an indication of oral anticoagulants of 92.1%, undertreatment of 24%, suboptimal control of vitamin K antagonists of 43%, use of antiaggregant as primary prevention of 13.42%, and primary health care monitoring of 34%. The implementation of activities aimed at improving the management of oral anticoagulants in the ACO-ZAR II study achieves a reduction in undertreatment up to 16%, in the use of antiaggregant up to 9%, and in suboptimal control up to 30%, as well as an increase in control from primary health care up to 69.2% and of the penetrance of direct oral anticoagulants up to 28%. (4) Conclusions: In conclusion, the application of activities aimed at optimizing the management of oral anticoagulants in health center patients allowed the improvement of risk assessment and registration, undertreatment, use of antiaggregant, suboptimal control of vitamin K antagonists, control by primary health care center, and the penetrance of direct oral anticoagulants.

## 1. Introduction

Atrial fibrillation (AF) is currently the most frequent sustained cardiac arrhythmia with an exponential increase with age [1], which will double figures in the next 50 years [2]. In the ATRIA study, [3] USA, and at Rotterdam [4] in Europe, the prevalence in the general population is approximately 1–2% and in our country, the REGICOR registries [5], CARDIOTENS [6], PREV-ICTUS [7], SPHINX [8], Val-FAAP [9], and OFFERS [10] reveal a similar scenario [11], with a prevalence of up to 4% in those over 40 years of age. Moreover, AF can also affect young adults or adolescents. Comorbidities such as hypertension, hyperthyroidism, or channelopathies may precipitate AF in the young [12,13,14,15]. Lifestyle factors such as endurance sport, alcohol consumption, or even smoking can also precipitate AF [16,17]. 

Alternatively, AF is independently associated with an increased risk of mortality for all causes 2 times higher in women and 1.5 in men [18,19]. It poses a high cardioembolic risk, increasing the risk of cerebrovascular accident (CVA) by 5 times, and is the leading cause of death in men and the second in women. This arrhythmia is also associated with increased morbidity in addition to thromboembolic events [20], such as heart failure, ischemic heart disease, or cognitive impairment. Regarding the great protagonist, stroke, AF is responsible for 20–25% of all of them [21,22], and these cardioembolic strokes are generally more severe, with higher mortality, and cause more residual disability.

AF is an important public health problem due to prevalence, morbidity, and mortality. However, AF appears in elderly patients that are polymedicated and with associated comorbidities, which makes it the prototype of a chronic-complex patient and a subsidiary of a comprehensive approach and multidisciplinary management [11].

Despite this important public health problem, only antithrombotic treatment is capable of modifying its natural history [23]. At present, the use of oral anticoagulation (OAC) in AF when indicated individually is inexcusable within a quality clinical procedure. The main scientific societies promote these treatment strategies, which consider thromboembolic prophylaxis essential in patients with AF [24,25,26], intending to improve their prognosis and quality of life.

In part, primary health care (PHC) has an enormous specific weight in the management of anticoagulation, with the percentage of patients diagnosed and followed up in our clinics [27,28] reaching 70% according to the latest national data, so we cannot avoid this responsibility.

Until recently, the standard treatment in oral anticoagulation was the use of dicoumarins or vitamin K antagonists (VKA), represented by acenocoumarol (Sintrom^®^) and Warfarin (Aldocumar^®^). However, one of the most important limitations of these drugs is the need for frequent monitoring of their interactions and their narrow therapeutic window [29]. To avoid these limitations, through randomized clinical trials, it has been shown that direct oral anticoagulants (DOACs) offer greater safety than dicoumarins for treating non-valvular AF (NVAF). Given these changes in anticoagulation treatments, it is necessary to adapt OAC management models, not only to encourage the adaptation of the professionals who prescribe them, but also for the health administration and the patients themselves who need these treatments [30,31].

These management systems must work beyond the clinical practice of professionals, adapting to the recommendations of the main scientific societies.

The Spanish Society for Healthcare Quality (SECA) [32] draws up a series of good practice recommendations for the safe use of DOACs in anticoagulated patients. These recommendations include the need for training activities for healthcare professionals, training, and information for patients with support unified and in writing, and the development of regulatory work procedures that reduce the variability of clinical practice.

According to the recommendations of the Spanish Agency for Medicines and Medical Devices (AEMPS) [33] in its Therapeutic Positioning Report (TPR) of 23 December 2013 (updated 21 November 2016), it recognizes that commercialized DOACs have shown a favorable risk–benefit ratio compared to VKAs in various clinical conditions and we are considering their prudent introduction.

In this line, we consider the ACO-ZAR I and II study to evaluate and improve the management of patients with atrial fibrillation in treatment with oral anticoagulants (OAC) from primary health care (PHC).

## 2. Materials and Methods

The ACO-ZAR project of “Improvement of oral anticoagulants the management in atrial fibrillation patients from primary health care center “Las Fuentes Norte”, as shown in Figure 1, began with the assessment of the starting situation through the ACO-ZAR I study. “Epidemiological characteristics and management of Atrial Fibrillation in primary health care (PHC)” defined the objectives for improving the management of the ACO. This is a cross-sectional descriptive study with clinical history data, developed from October 2015 to March 2016, including the entire population assisted with AF in the 434 patients of the 22,513 users, of whom 49 were lost for various reasons, maintaining a final sample of 385 patients.

After performing the ACO-ZAR I descriptive study, the ACO-ZAR II study of “Optimisation of oral anticoagulants in primary health care”, in the case of a prospective quasi-experimental study, was carried out on the previous population of the ACO-ZAR I study (385 patients with AF), with the loss of 67 patients due to a lack of follow-up (displacement or death), in the case of a final sample of 318 patients, from June 2016 to July 2017.

In this study, the situation of the oral anticoagulants is reassessed by repeating the methodology of the previous study, but after performing activities aimed at patient information in a unified and written format, training of health personnel through clinical sessions and protocolization of care activity can be achieved through the development of quick guides shown in Appendix A. Additionally, action protocols were established for the different therapeutic options, which reduces the variability of clinical practice. All this information has been based on the recommendations of main scientific societies [32,33].

The sources of information were the medical records of our electronic health record and the electronic prescription. The main variables are the existence of atrial fibrillation (FA) diagnosis in electronic health record. The diagnosis is defined using the International Classification of Primary Care (ICPC-3) [34]. Given the diversity of nomenclatures and registration in the computer system for the same diagnosis, the episodes included among the participants were agreed upon. For open a new atrial fibrillation diagnosis, it is necessary that the patient be documented through the performance of an electrocardiogram that documents the presence of atrial fibrillation [35]. 

This electronic health record includes sustained and non-sustained atrial fibrillation. The CHADS2-VASC scale [36,37] was used to assess the thromboembolic risk and the suitability of the OAC treatment indication, and the HAS-BLED scale [38,39] was used to assess the hemorrhagic risk, both included in our computer support. To evaluate the place and degree of control in patients receiving VKA treatment, we used the time in therapeutic range (TTR). Correct VKA treatment was considered optimal if a TTR > 60% in the last 6 months, as required by the IPT. Blood analytical data evaluated the renal and liver function. The parameters evaluated in the blood test were glomerular filtration rate (GFR), bilirubin, glutamic oxaloacetic transaminase (GOT), glutamic-pyruvic transaminase (GPT), and alkaline phosphatase (ALP) from recent controls requested from PC or Specialized Care (SC), according to the EHR record. Renal failure was considered if GFR < 60 mL/min/1.73 m^2^ was calculated from the MDRD-4 equation [40]. 

Liver failure was considered if one of these 3 situations appeared (Bilirubin values greater than twice the maximum normal levels; GOT, GPT, or ALP levels greater than three times upper normal levels; open a new episode of Hepatopathy associated with coagulopathy with a clinically relevant risk of bleeding: cirrhosis). Associated treatment data (antiaggregants, mon-steroidal anti-inflammatory drugs (NSAIDs), drugs for heart rate control or rhythm control) were obtained from electronic prescription RE.

Ethical considerations: The ACO-ZAR project “Optimization of oral anticoagulants in primary health care” was included in the Management Agreement/Program Contract of the Aragonese Health Service 2016–2017, as shown in Appendix A. The study was conducted out following the principles of the Declaration of Helsinki and Tokyo. 

Statistical analysis: The statistical package SPSS version 25 was used. To observe the type of distribution of variables used, the Kolmogorov–Smirnov test with Lilliefors corrections for describing qualitative variables were used for frequencies and percentages. For describing quantitative variables, means, standard deviations, and ranges were used in the case of normal behavior, as well as the medians and interquartile ranges for the rest of the cases. The respective 95% confidence intervals were obtained.

## 3. Results

The ACO-ZAR I study revealed that the population with AF treated in our center had a mean age of 76.58 ± 10.62 (with ≥75 years 68%), with a prevalence of 1.7% in the general population, which increased exponentially with age, assuming 2.98% of those more than 40 years old had an abundance of associated comorbidity, mainly arterial hypertension (76.3%), diabetes mellitus (35.2%), chronic renal failure (32.3), heart failure (27.9%), thromboembolic disease (18.8%), and a high degree of polypharmacy (84.5%). Table 1 shows the relationship between the different comorbidities and the sex of the participants.

### 3.1. Thromboembolic and Hemorrhagic Risks

The risks—thromboembolic due to CHADS2-VASC and hemorrhagic due to HAS-BLED—did not follow a normal distribution (*p* < 0.05%) and were moderate-to-high (CHADS2-VASC median 4, IQ interval 2–6 and HAS-BLED median 3, IQ interval 1–5). CHADS2-VASC increased statistically significantly with age and HAS-BLED showed statistically significant differences depending on the age range. 

Concerning thromboembolic risk, 91.93% of the population had a CHADS2-VAS C ≥ 2, 4.43% had moderate thromboembolic risk by CHADS2-VAS C = 1, and 3.65% did not have thromboembolic risk. 

Alternatively, 57% had a high bleeding risk (HAS-BLED ≥ 3).

Regarding the management of OAC treatment, only 10.5% of patients had an evaluation of thromboembolic risk (CHADS2-VASC) and hemorrhagic risks (HAS-BLED), with 24% undertreatment. Additionally, 13.42% used antiaggregant AAG as primary prevention, 43% had suboptimal control of VKA, 22.10% used DOAC, and 34% was controlled exclusively from the primary health care center.

Finally, regarding symptomatic treatment, 68.8% were in rate control and 29.9% were in rhythm control. None of the therapeutic options presented significant differences due to age or associated pathologies. 

After the correlational analysis, we can observe how the thromboembolic risk increases slightly as the patient gets older, and as it happens when the patient is a woman. Additionally, the thromboembolic risk increases when the patient is polymedicated and decreases when the bleeding risk is lower than 3 points. These data are shown in Table 2 and Table 3.

Alternatively, we can observe how the hemorrhagic risk increases slightly as the patient gets older. Additionally, the hemorrhagic risk decreases when the patient does not have a habitual treatment with antiaggregants or NSAIDs. These data are shown in Table 2 and Table 4.

### 3.2. Management of Oral Anticoagulants in Primary Health Care Center

The results of ACO-ZAR II, after implementation of activities aimed at optimizing the management of OAC in primary health care patients, showed an evaluation and registration of risks of 35.5%; undertreatment of 16%; use of antiaggregant drugs of 9%; suboptimal control of VKA of 30%; control by primary health care center of 69.2%; penetrance of DOAC of 28%; and uptake of the follow-up of DOAC in nursing management of 21.9%. A comparison of the results obtained between the assessment of ACO-ZAR I and those obtained for ACO-ZAR II is shown in Figure 2.

## 4. Discussion

The epidemiological results of ACO-ZAR I revealed a frail population due to advanced age, multiple associated comorbidities and polypharmacy, representative of the populations studied at the national level, an important fact when extrapolating our management data with studies with a broader scope than the of the primary health care center itself and national nature, due to the lack of other similar studies of Aragon.

Regarding the management of OAC, a low evaluation of thromboembolic and hemorrhagic risks was observed by CHADS2-VASC and HAS-BLED, respectively, of only 10.5%, of which no national data are available in this review. Regarding our undertreatment of 24%, we were at the national level (ranging from 16% in the FIATE study [27] to 23.5% in the AFABE study [41], the same as in the percentage of suboptimal control of 43% (national data between 40–50% according to the series), but in the exclusive control by primary health care with 34%, we were well below the national data ranging from 59.6% from FIATE (46.8% in Aragón) to 70% from PAULA [28], although comparable to the results of the AGORA project [42].

The optimization objectives of the OAC emerged for our population and the activities to achieve this were through the ACO-ZAR II study. The implementation of activities aimed at optimizing the management of OAC in primary health care patients showed an increase in risk assessment and registration from 10.5% to 35.5 and decrease in undertreatment from 24% to 16%, matching the result of the FIATE study [27]. Moreover, decrease in the use of antiaggregant as primary prevention of stroke, going from 13.42% to 9% in line with the Spanish cardiology Society (SEC), recommends excluding antiaggregant as an alternative to OAC in the prevention of stroke and systemic embolism in AF, below the ESCONDIDA study [43]. Moreover, evidence from controlled trials supports the appropriate prescription of OACs for preventing CVA [44].

Furthermore, a decrease in suboptimal control from 43% to 30% and an increase in the penetrance of DOACs from 22% to 28% placed us within the national average, and the increase in control by primary health care center from 34% to 69.2% was in line with the results of the PAULA study [28].

A study strength is the use of the CHADS2-VAS C and HAS-BLED scales for a complete joint evaluation of thromboembolic and hemorrhagic risks, which to date has not been carried out in previous studies. Additionally, this type of study had not been carried out in the Aragonese population, and therefore, data on the implementation and follow-up of atrial fibrillation were not known. Finally, it is important to bear in mind that very few studies have been conducted to date looking for this perspective of the treatment of atrial fibrillation from primary health care, this being the main and first contact related to the health of patients and that with which patients have the most contact.

Meanwhile, in terms of the limitations of performing this study, note that the sample was obtained only from a health center. However, in the sampling, the entire affected population has been selected, observing population characteristics with similarities to those of the national studies, allowing a viable comparison with the results of these. It is also necessary to insist on the existence of a bias in the results, in terms of suboptimal control and follow-up percentage in primary health care. This bias is due to the imposition of sector II to assume from primary health care center the patients followed in Hematology of the Hospital Universitario Miguel Servet already stabilized, according to the criterion of 3 consecutive INR in range as of March 2017.

## 5. Conclusions

In conclusion, the population of our study has a moderate-high risk of thromboembolism (according to CHADS2-VASC) and high hemorrhagic risk (according to HAS-BLED). Additionally, the percentage of undertreatment as suboptimal control, and the lack of evaluation and management, is also high. For this situation, the application of the activities aimed at optimizing the management of OAC in primary health care patients allowed the achievement of the proposed objectives, improving the evaluation and registration of risks, undertreatment, and the use of antiaggregant, suboptimal control of VKA, and control by primary health care center and DOAC penetrance.

Observing the evidence and results obtained, it is increasingly necessary to give importance to AF in general and thromboembolic and hemorrhagic events. CHADS2-VAS C and HAS-BLED scales enable a complete joint evaluation of thromboembolic and hemorrhagic risks to allow a notable improvement in the quality of care and the quality of life of the population with AF. 

Given the importance of primary health care in clinical practice, the quality of oral anticoagulation, and the changes in the supply of anticoagulant treatment, there is a need to adapt the management model and improve the information provided to the patient and therapeutic adherence to OAC users. These improvements are promoted through the creation of training programs and better action protocols that allow the reorganization of daily clinical activity for the different therapeutic options.

More scientific evidence is needed when applying new OAC optimization and management protocols, in addition to considering the possible long-term effects of these management models.

## Figures and Tables

**Figure 1 ijerph-19-06746-f001:**
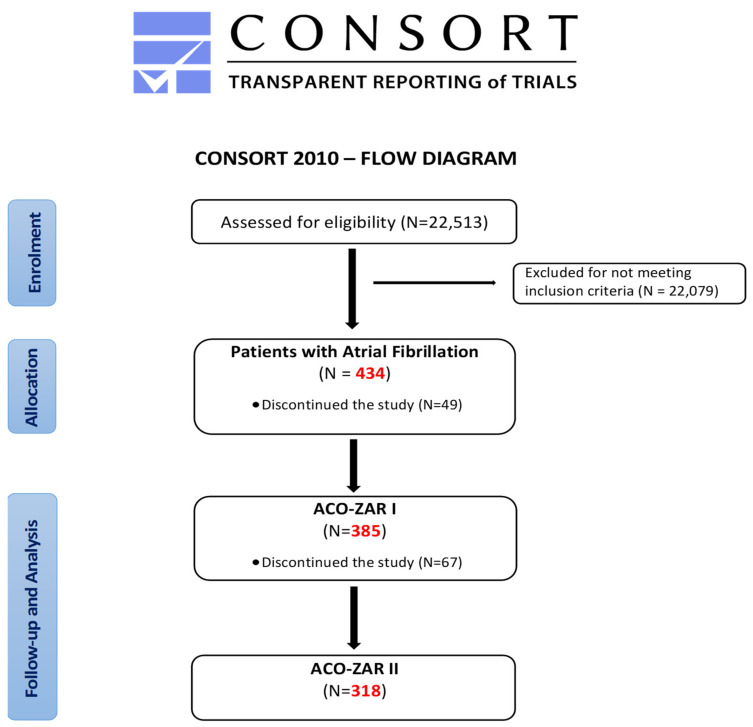
Consort Flow Diagram.

**Figure 2 ijerph-19-06746-f002:**
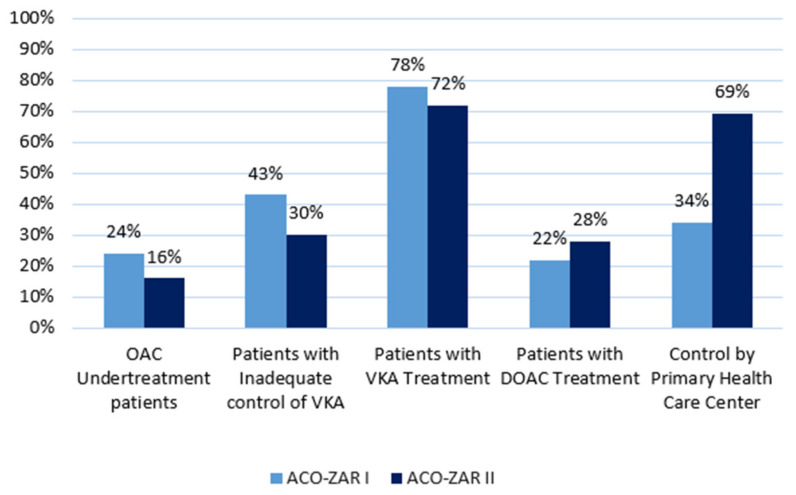
A comparison of the results obtained between ACO-ZAR I and ACO-ZAR II.

**Table 1 ijerph-19-06746-t001:** Cross-tabulation of variables with gender.

	Female		Male		
Variables	*n*	Med. (IQR)/% (95% CI) *	*n*	Med. (IQR)/% (95% CI) *	*p*-Value
Age	202	79 (74–84)	182	78 (68–83)	0.258
Arterial hypertension	169	57.68 (51.97–63.24)	124	42.32 (36.76–48.03)	*p* < 0.001
Diabetes Mellitus	83	61.48 (53.10–69.38)	52	38.52 (30.62–46.90)	0.014
Chronic renal failure	80	63.49 (54.86–71.52)	46	36.51 (28.48–45.14)	0.004
Chronic liver failure	0	0.00	2	100.00	0.101
Heart failure	65	60.75 (51.31–69.91)	42	39.25 (30.39–48.69)	0.052
Stroke CVA	34	50.00 (38.32–61.68)	34	50.00 (38.32–61.68)	0.688
Peripheral Arterial Disease	17	40.48 (26.67–55.54)	25	59.52 (44.46–73.33)	0.103

*n*: absolute value; med.: median; IQR: interquartile range; %: percentage; CI: confidence interval. * expressed as % (95% CI); *p*-value: statistical significance; CVA: cerebrovascular accident.

**Table 2 ijerph-19-06746-t002:** Correlation of variables with CHADS2-VASC and HAS-BLED.

	CHADS2-VASC		HAS-BLED	
Correlation between Variables	*p-*Value	r	*p-*Value	r
Age	*p* < 0.001	0.576	*p* < 0.001	0.571
Gender	*p* < 0.001	0.463	0.246	0.059
Polypharmacy	*p* < 0.001	0.265	0.014	0.126
Alcohol consumption	*p* < 0.001	−0.252	0.275	0.056
Bleeding	0.440	0.40	*p* < 0.001	0.287
Treatment with NSAIDs	0.307	−0.052	*p* < 0.001	0.164
Treatment with Antiaggregants	0.216	0.064	*p* < 0.001	0.330
CHADS2-VASC	-	-	*p* < 0.001	0.519
Absence of registration CHADS2-VASC	0.699	−0.020	0.315	−0.052
HAS-BLED.	*p* < 0.001	0.519	-	-
The absence of registration HAS-BLED.	0.699	−0.020	0.315	−0.052

*p*-value: statistical significance; r: correlation coefficient; NSAIDs: Non-steroidal anti-inflammatory drugs.

**Table 3 ijerph-19-06746-t003:** Cross-tabulation of variables with CHADS2-VASC.

Variables	Coefficient	*p-*Value	Importance
Intercept	−0.670	0.268	
Gender (Male)	−1.169	*p* < 0.001	0.410
Age (Older)	0.055	*p* < 0.001	0.232
HAS-BLED (≤3 points)	−0.868	*p* < 0.001	0.171
Polypharmacy	1.056	*p* < 0.001	0.168
No Alcohol consumption	0.450	0.040	0.020

*p*-value: statistical significance.

**Table 4 ijerph-19-06746-t004:** Cross-tabulation of variables with HAS-BLED.

Variables	Coefficient	*p-*Value	Importance
Intercept	0.429	*p* < 0.001	
Age (Older)	0.059	*p* < 0.001	0.270
No treatment with Antiaggregants	−1.123	*p* < 0.001	0.270
No Bleeding	−1.332	*p* < 0.001	0.215
CHADS2-VASC (High score)	0.245	*p* < 0.001	0.141
No treatment with NSAIDs	−1.059	*p* < 0.001	0.092
No Polypharmacy	−0.290	0.043	0.012

*p*-value: statistical significance. NSAIDs: Non-steroidal anti-inflammatory drugs.

## Data Availability

The data presented in this study are available on request from the corresponding author.

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
