# Peer review of "Improvement in the Management of Oral Anticoagulation in Patients with Atrial Fibrillation in Primary Health Care"

_ijerph, 2022, doi:10.3390/ijerph19116746_

Round 1

Reviewer 1 Report

Dear Authors,

Congratulations on the prepared manuscript.

Please specify the risk scale used for the occurence of thromboemolic complications in patients with atrial fibrillation. Is the scale used CHA2DS2-VASc oraz CHADS2. This is essential for risk assessment.

In the text and tables instead of correct HAS - BLED, an incorrect HAS - BELD was used.

Best regards

Author Response

Thanks for the contributions, the scale that has been used for both the CHA2DS2-VASc and HAS-BELD has been specified in the methodology, and the citations referring to said scales have been incorporated into the bibliography.

Regarding the results, thanks for the contribution, the HAS-BLED risk variables have been corrected and specified both in the text and in the tables.

Reviewer 2 Report

Thank you for submitting your paper "Improvement of treatment control with oral anticoagulants in Primary Care: A pending account".  Based on your analysis, it appears the educational intervention and implementation of protocols improved prescribing of appropriate anticoagulation in a population of patients with atrial fibrillation.  I recommend making changes that would improved the presentation of your information.  

  1. The English grammar could use extensive editing to ensure the information you aim to communicate is translated correctly.  Many sentences are too long and do not have the required punctuation to convey the information I believe you hope to present.  Enlisting the assistance of an editorial assistant who is proficient in translating and writing English would be helpful.
  2. Many abbreviations are used throughout the manuscript that are not universally accepted.  If these are not needed, it would help the reader if the entire term was used.  If felt the abbreviations are preferred, please provide definitions (FA, AP, RE, ACOD, CS), others are defined differently in other countries.  For example, GRF is abbreviated in the US as GFR.
  3. The title is a bit confusing.  I believe this is really Improvement in management of oral anticoagulation in patients with atrial fibrillation in primary care. I believe you are describing the benefits of an educational initiative that includes clinical training and protocol  implementation to improve prescribing and implementation of appropriate oral anticoagulation.  The title could be improved to describe this analysis.

Author Response

Thank you very much for your contribution, grammatical and punctuation corrections have been made so that the information in the article is better understood by the reader. The use of abbreviations has also been reduced and of those that are relevant, those that may have universally unified use have been reviewed.

About the title, I consider your contribution very relevant, and I mention it as an addition to the title of the article.

Reviewer 3 Report

the work entitled "Improvement of treatment control with oral anticoagulants in Primary Care: A pending account." It is very well developed and carried out in a health center. The application of the activities aimed at optimizing the management of OAC in CS patients allowed the achievement of the proposed objectives, improving the evaluation and registration of risks, undertreatment, the use of AAG, suboptimal control of VKA, control by PA and DOAC penetrance. It is a very interesting work given the prevalence of this entity today, and with the greater life expectancy of the population. I consider that the work should be accepted for publication without corrections

Author Response

Thank you very much for your interest and your positive contributions to our work

Reviewer 4 Report

I would congratulate with authors for this very good paper. This topic is extremely interest since oral anticoagulant prescription for stroke prevention in atrial fibrillation patients frequently does not follow current guidelines with underuse in patients at high risk of stroke and substantial overuse in those at low risk. Here you find comments in order to improve the manuscript:

-Introduction (line 40): “with a prevalence of up to 4% in those over 40 years of age” . While mostly seen in elderly, authors should explain that AF can also affect young adults or adolescent. In particular, AF in the young may be precipitated by hypertension, hyperthyroidism (doi:10.1001/archinte.164.15.1675), lifestyle factors such as endurance sport (https://doi.org/10.1093/europace/eun289), alcohol consumption, even smoking (doi: 10.1016/j.hrthm.2011.03.038), and finally channelopathies (doi: 10.1111/jce.14787 ; 10.1161/CIRCEP.119.007213 ;  doi: 10.1111/jce.13410). Please cite this very important points, including all six suggested references.

- Methods: How the AF was documented (ECG?, Holter? Implantable loop recorder?). This topic is extremely important in order to potentially better determine asympatomatic AF patients. In this scenario it would be also relevant to distinghish between sustained and non sustained (paroxysmal) AF forms

- Results: How the authors considered a  patient with CHADS-VASC score = 1, representing today a grey area in population with an AF burden. Please explain

- Results : 29.9% of patients were in rhythm control. Did any patient underwent to AF ablation? This may impact also on the management of OAC in younger patients with a low CHADS-VASC score

- Discussion:  an  important reference are missing in the discussion, including a nice systematic review from Pritchett et al (doi: 10.1055/s-0038-1676835) of controlled and uncontrolled studies with interventions designed to improve appropriate OAC prescription for stroke prevention in eligible AF patients (according to risk assessment tool or guidelines): the main outcome was change in proportion of eligible AF patients prescribed OACs for stroke prevention

- Conclusions: authors should summarize conclusions since too long, integrating part of conclusions in the discussion section

Author Response

Thank you very much for your interest and your contributions to our work.

It has been incorporated into the introduction of the manuscript and the referenced citations.

It has been incorporated in detail how the patients with AF were selected and through which system this diagnosis was recorded in the methodology section.

The atrial fibrillation was defined by the existence of an open episode in electronic health record using International Classification of Primary Care (ICPC-3). For open a new atrial fibrillation diagnosis, it is necessary the patient is documented through the performance of an electrocardiogram that documents the presence of atrial fibrillation. In this electronic health record includes sustained and non-sustained atrial fibrillation. Therefore, in this study, it has not been included as a differentiating variable to be studied. It is true that it is a relevant fact that we will apply in later studies.

As suggested in results, the percentages of the participating population have been incorporated according to CHADS-VASC risk equal to 0, moderate or equal to 1, and high risk greater than 2.

In our case, it is a very low percentage (around 4%), so it does not allow a comparative analysis with a high risk or with no risk, if the population itself is a bias. Therefore, when incorporating the correlations and the multivariate analysis, we incorporated the CHADS-VASC as a quantitateive variable and not separated by the 3 risk classifications, allowing the incorporation of the CHADS-VASC=1 to incorporate it at the same level as the CHADS-VASC > 2.

Thank you for your contribution. Ablation treatment of atrial fibrillation was not considered as a variable to be collected in the study. It was decided through the research team that considering the age of the population of the health area, that it was not the treatment of frequent choice in this population age. It is true that for the following investigations it will be a relevant variable to consider.

Round 2

Reviewer 4 Report

The manuscript definitely improved. I have no further comments. Congratulations to the authors

Author Response

Thank you very much for all the contributions you gave to this study. Your review and interest have motivated us to further improve the evidence,

sincerely